# Slow Learning and Fast Inference: Efficient Graph Similarity Computation via Knowledge Distillation

**Can Qin**[1*]   **Handong Zhao**[3]   **Lichen Wang**[1]   **Huan Wang**[1]   **Yulun Zhang**[1]   **Yun Fu**[1,2]

[1]Department of Electrical and Computer Engineering, Northeastern University
[2]Khoury College of Computer Science, Northeastern University
[3]Adobe Research

## Abstract

**Graph Similarity Computation (GSC)** is essential to wide-ranging graph applications such as retrieval, plagiarism/anomaly detection, etc. The exact computation of graph similarity, e.g., Graph Edit Distance (GED), is an NP-hard problem that cannot be exactly solved within an adequate time given large graphs. Thanks to the strong representation power of graph neural network (GNN), a variety of GNN-based inexact methods emerged. To capture the subtle difference across graphs, the key success is designing the dense interaction with features fusion at the early stage, which, however, is a trade-off between speed and accuracy. For **Slow Learning** of graph similarity, this paper proposes a novel early-fusion approach by designing a co-attention-based feature fusion network on multilevel GNN features. To further improve the speed without much accuracy drop, we introduce an efficient GSC solution by distilling the knowledge from the slow early-fusion model to the student one for **Fast Inference**. Such a student model also enables the offline collection of individual graph embeddings, **speeding up the inference time in orders**. To address the instability through knowledge transfer, we decompose the dynamic joint embedding into the static pseudo individual ones for precise teacher-student alignment. The experimental analysis on the real-world datasets demonstrates the superiority of our approach over the state-of-the-art methods on both accuracy and efficiency. Particularly, we speed up the prior art by more than 10x on the benchmark AIDS data.

## 1   Introduction

Measuring the similarity across graphs, i.e., **Graph Similarity Computation (GSC)**, is one of the core problems of graph data mining, centered around by multiple downstream tasks such as graph retrieval [1, 2], plagiarism/anomaly detection [22, 41], graph clustering [39], etc. As shown in Fig. 1, the graph similarity can be defined as distances between graphs, such as **Graph Edit Distance (GED)**. The conventional solutions towards GSC are the exact computation of these graph distances, which, however, is an NP-hard problem. Therefore, such exact solutions are less favorable when handling large-scale graphs due to the expensive computation cost. Computational time, especially run time in inference stage, is particularly important in industrial scenarios. As a motivating example, in graph-structured molecules or chemical compounds query for in-silico drug screening, fast identifying similar compounds in a large database is a key process [25].

Leveraging the strong representational power of graph neural network (GNN) [21, 13, 43, 42], the GNN-based approximate GSC solutions have gained increasing popularity. To adapt GNNs to the GSC task, the target similarity score (e.g., GED) is normalized into the range of (0, 1]. In this way, the

---

*Corresponding Author: `qin.ca@northeastern.edu`

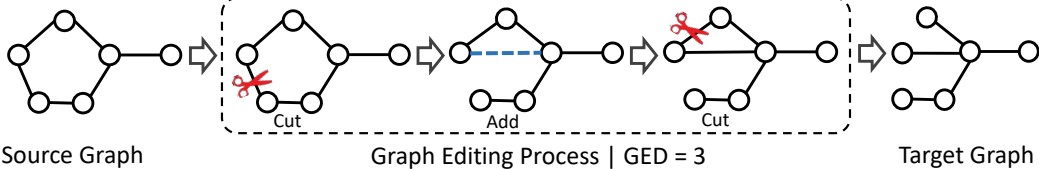

Source Graph    Graph Editing Process | GED = 3    Target Graph

Figure 1: Illustration of graph edit distance (GED), which is defined as the number of edit operations in the optimal path to transform the source graph to the target graph.

GSC can be regarded as a single-value regression problem that outputs a similarity score given two graphs as inputs. A standard design can be summarized as a twin of GNNs bridged by a co-attention with a Multi-layer Perceptron (MLP) stacked as the regression head. Such approaches can be trained in a fully supervised way using the Mean Square Error (MSE) loss computed over the ground truth similarity score. Many GNN-based GSC methods [1, 2, 22] followed such strategy, which, however, suffers from the fusion issue.

The paper presents a novel solution to both effectively and efficiently address the task of approximate GSC. Compared to the commonly used graph convolutional network as the backbone [1, 2], this paper adopts a more robust network, i.e., Graph Isomorphism Network (GIN) [43]. Cross-graph fusion is essential to the model. The multi-scale features within different GIN layers are fused with a new design. We have adopted an attention layer stacked over the concatenated cross-graph features for smooth feature fusion. To this end, similar features will be assigned with more weights to contribute to the desired task. Moreover, to make the model easier to deploy, we take an MLP for feature learning which is simple but effective to achieve cutting-edge performance.

Intuitively, speed and accuracy can be considered as a trade-off. GSC naturally requires dense connections/interactions between the two input graphs, which will consequently cause increasing computations as the cost. This paper focuses on the efficiency of inference speed which can be addressed by either model compression or a faster data loading pipeline. Especially in industrial scenarios, the raw graph data are usually pre-processed as the embeddings off-line that can be easily applied to the real-time downstream tasks, e.g., molecular graph retrieval. However, as shown in Fig. 2, most of the co-attention-based GSC solutions employ feature fusion in the early stage, which only outputs the joint embedding of pairing graphs. Inspired by [26],

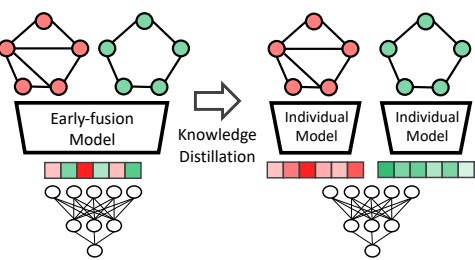

Figure 2: Illustration of knowledge distillation to achieve a fast model (right side) given a early-fusion-based slow model (left side).

we propose a lightweight model that removes all the early feature fusion modules in the encoder for efficient GSC. In this way, as shown in Fig. 2, the individual embedding of each graph can be collected by a Siamese GNN. Such pairing graph embeddings will be fused with an attention layer to predict the final similarity score.

To overcome the accuracy drop of such a small network, we take a novel paradigm of **Knowledge Distillation (KD)** specifically designed for our task. As shown in Fig. 3, we propose an early-feature fusion network regarded as the teacher model, and the student model is a siamese network without co-attention. It is found that the direct distillation of joint embeddings fails to work where the KD loss disturbs largely during training. To solve this, we generate the pseudo individual embeddings of the teacher model and use them for KD by minimizing their relational distances [29]. To ensure pseudo individual embeddings fully cover the information of raw graphs, we further apply an MSE loss on the reconstructed joint embeddings concatenated from pseudo individual ones. We have verified that there is only a marginal accuracy drop compared with the original joint embeddings, which justifies the claim above. To sum up, our contributions can be summarized in three folds:

- We introduce a new early-feature fusion model to achieve the competitive accuracy by designing a strong co-attention network and taking the GIN as the backbone.

- For efficient inference and off-line embedding collection, we propose a novel Knowledge Distillation method for GSC where the joint embeddings are decomposed to distill.

- Extensive experiments on the popular GED benchmarks demonstrate the superiority of our model over the state-of-the-art GSC methods on both accuracy and efficiency. Compared with the co-attention models, there is a more than **10 times faster** in inference speed compared with the best competitor on AIDS dataset. [†]

## 2 Related Works

### 2.1 Graph Similarity Computation (GSC)

Graph similarity computation measures the similarity of two given graphs, where similarity metrics can be defined as Graph Edit-Distance [6], Graph Isomorphism [8], and Maximum Common Subgraph [7]. Exact computation of these metrics is generally an NP-complete problem [48]. To speed up the computation, kernel-based methods have been extensively proposed to approximate the exact solvers [44, 3, 28, 45]. Recently, inspired by the strong representation power of deep neural network, a number of neural network based methods have been proposed and demonstrated a huge success [47, 1, 2, 24, 22, 41, 40]. Among them, regression-based similarity learning has a great promise due to the competitive performance in both efficiency and efficacy [1, 22, 2]. The intuition here is to learn an embedding vector using a graph neural network (GNN), and then measure the similarity of graph embeddings. While such a graph-level embedding encoded by GNN alone is not sufficient to well distinguish the nuances of subgraph level structures. To integrate subgraph information for final similarity computation, several methods are proposed recently, such as node-level pairwise comparison [1], cross-graph attention-based matching [22], multi-scale neighbor aggregation [2], etc. Despite the superior efficacy reported under various metrics (such as Accuracy, Mean Squared Error (mse), Spearman's Rank Correlation Coefficient), the complex subgraph matching/fusion components (termed 'early-fusion' in Fig. 2) in different layers dramatically slow down the similarity measure. Moreover, early-fusion prevents pre-computing the embeddings for all candidate graphs for further reducing inference time in the graph retrieval scenario. Motivated by this, we propose a slow learning and fast inference method by leveraging the knowledge distillation idea to transfer the fine-grained but slowly learned early-fusion teach model to the fast-inference student model.

### 2.2 Knowledge Distillation (KD)

Knowledge distillation is a general neural network training method, where a (typically pretrained) teacher network is introduced to guide the learning of a student network. Its idea was first pioneered by Bucilua et al. [5] to compress large machine learning models, where they proposed to transfer the knowledge of a model ensemble into a neural network by labeling unlabeled data as transfer set. This idea was later refined by Hinton et al. [16], where they adopted softened probabilities of the teacher as a target for the student to learn and coined the term "knowledge distillation". Ever since, many methods have been proposed revolving around the central question in KD: "what is the definition of knowledge to be distilled". Popular definitions include feature distance [34], feature map attention [46], feature distribution [30], activation boundary [15], inter-sample distance structure [29, 32, 23, 36], and mutual information [35]. See [38, 11] for a more comprehensive survey. [26] is proposed to distill separate models from a co-attention one. Despite the progress, they mainly focus on convolutional neural networks for vision tasks (mainly image recognition) or recurrent neural networks for sequential data tasks (e.g., for natural language understanding [19]).

## 3 Approach

This section will introduce 1) the architecture of the early-fusion network (i.e., teacher model); 2) the KD process and its interpretation. Before that, we start from the formalized problem definition.

### 3.1 Problem Formulation

Formally, a graph $\mathcal{G}$ is defined upon the node set $\mathcal{V}$ and edge set $\mathcal{E}$ as $\mathcal{G} = (\mathcal{V}, \mathcal{E})$. In specific, the edge linking a pair of nodes including $u \in \mathcal{V}$ and $v \in \mathcal{V}$ can be denoted as the $(u, v) \in \mathcal{E}$. In our setting, all the accessible graphs are undirected, i.e., $(u, v) \in \mathcal{E} \leftrightarrow (v, u) \in \mathcal{E}$. The quantity of nodes is represented as $N = |\mathcal{V}|$. A convenient way to represent the graphs is the adjacency matrix

---

[†]The code is uploaded on https://github.com/canqin001/Efficient_Graph_Similarity_Computation

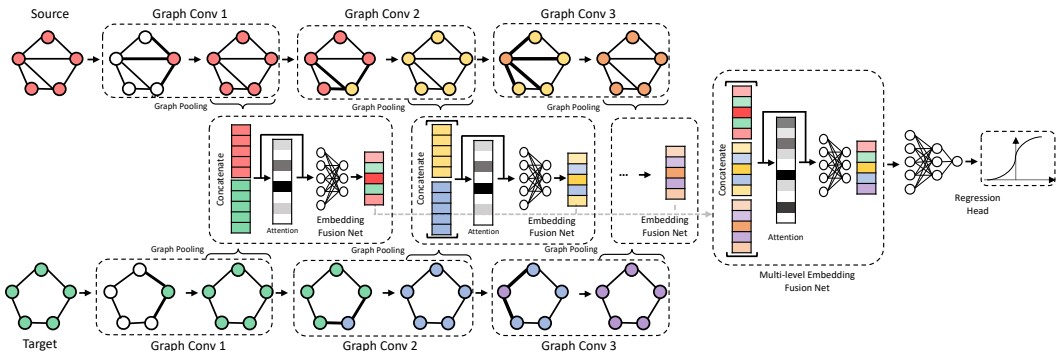

Figure 3: Overview of early-feature fusion network (Teacher Net) which is composed of a feature encoder and a regression head as the whole. Within the the feature encoder, there are multiple components including GIN as the backbone, the Embedding Fusion Network (EFN) and graph pooling. The regression head is a MLP which projects the joint embedding into the desired similarity.

$\mathbf{A} \in \mathbb{R}^{|\mathcal{V}| \times |\mathcal{V}|}$. We denote the presence of edges as $\mathbf{A}[u, v] = 1$ if $(u, v) \in \mathcal{E}$ and $\mathbf{A}[u, v] = 0$ otherwise. Mostly, graph attributes (e.g., node labels) are available. Such node-level features can be denoted as a real-value matrix $\mathbf{X} \in \mathbb{R}^{|\mathcal{V}| \times m}$ with the $m$ dimension and the order of feature matrix $\mathbf{X}$ is consistent with the adjacency matrix [12].

In GSC task, we have the access to pairing graphs $\mathcal{G}_i$ and $\mathcal{G}_j \in \mathcal{D}$, where $\mathcal{D} = \{\mathcal{G}_0, \mathcal{G}_1, ...\}$ is the graph set. The similarity of such two graphs can be represented as Graph Edit Distance (GED) or Maximum Common Subgraph (MCS). As shown in Fig. 1, the GED is defined as the number of edit operations in the optimal trajectory to transform the source graph to the target. The MCS is the maximum subgraph common to both two graphs. To well fit GNN, the standard GED value is normalized as the nGED, i.e., $\text{nGED}(\mathcal{G}_i, \mathcal{G}_j) = \frac{\text{GED}(\mathcal{G}_i, \mathcal{G}_j)}{(|\mathcal{G}_1| + |\mathcal{G}_2|)/2}$. In the following, nGED should be transformed to the value ranging (0, 1] as the ground truth similarity score $s_{ij}$, i.e., $s_{ij} = exp(-\text{nGED}(\mathcal{G}_i, \mathcal{G}_j)) \in \mathbf{S}$, where $\mathbf{S} \in \mathbb{R}^{|\mathcal{D}| \times |\mathcal{D}|}$ indicates the similarity matrix among all the graphs [2] .

## 3.2   Early-fusion Network (Teacher Model)

As discussed above, the key success of GSC is to enrich the interaction between the pairs of graphs through feature extraction. Therefore, our teacher model follows the conventional approaches [1, 2, 22] that fuse the cross-graph features in the early stage. The architecture of our proposed early-fusion (teacher) model is shown in Fig. 3. Specifically, we take the Graph Isomorphism Network (GIN) [43] as the backbone model for abstract feature extraction. The multi-level features are encoded within different convolution layers. For smooth fusion, we take an attention layer to enrich the representation ability of the embeddings and take an MLP for further feature learning. More details are given below.

### 3.2.1   Graph Isomorphism Network (GIN)

The isomorphism on graphs, i.e., $\mathcal{G}_i \simeq \mathcal{G}_j$ , is defined as a bijection between $\mathcal{G}_i$ and $\mathcal{G}_j$: $f : V(\mathcal{G}_i) \to V(\mathcal{G}_j)$. Graph isomorphism is highly related to GSC where the graphs isomorphism also represents that the GED is 0: $\mathcal{G}_i \simeq \mathcal{G}_j \leftrightarrow GED(\mathcal{G}_i, \mathcal{G}_j) = 0$. Therefore, the strong power of GIN in representing the graph isomorphism will be beneficial to GSC. GNN involves multiple learning steps, including message passing, node feature updating, and readout. Let $\mathcal{A} : \mathcal{G} \to h \in \mathbb{R}^d$ denote a general GNN. The iterative updating of node features from the $(k-1)$-th to the $k$-th layer can be formulated as:

$$h_v^{(k)} = \phi \left( h_v^{(k-1)}, f(\{h_u^{(k-1)} : u \in \mathcal{N}(v)\}) \right), \tag{1}$$

where $\mathcal{N}(v)$ is the set of neighbouring nodes of node $v$ and its embedding at layer $k$ is denoted as $h_v^{(k)}$. $\phi$ and $f$ represent the different mapping functions. In GIN, it has been discussed that the MLP can model the $f$ and $\phi$ very well due to the universal approximation theory [18, 17]. Therefore, the composition of $f^{(k+1)} \circ \phi^{(k)}$ is replaced by an MLP. The node embedding of GIN is updated as:

$$h_v^{(k)} = \text{MLP}^{(k)} \left( (1 + \epsilon^{(k)}) \cdot h_v^{(k-1)} + \sum_{u \in \mathcal{N}(v)} h_u^{(k-1)} \right), \tag{2}$$

where $\epsilon^{(k)}$ can be either learnable or fix parameter. To readout the graph's global embedding, multiple order-invariant mapping functions, such as 'mean', 'max' or 'sum', are useful for information aggregation. In GIN, it has been verified that 'sum' is the most powerful one to learn and model all the labels without the constraints of node quantities. Therefore, GIN takes the 'sum' as the aggregator:

$$h_{\mathcal{G}} = \text{CONCAT}\left(\left(\text{sum}(\{h_v^{(k)}|v \in \mathcal{G}\})|k=0,...,K\right),\right. \tag{3}$$

where the features in all the layers, i.e., from layer 0 to layer $K$, are concatenated as the global feature. In this paper, we take $K$ as 2 where there are 3 GIN layers in total for feature learning.

### 3.2.2 Embedding Fusion Network (EFN)

Feature fusion across graphs is crucial for GSC. In this paper, we have proposed a novel **Embedding Fusion Network (EFN)** as part of the whole framework to address such a challenge. The inputs fed into EFN are graph-level embeddings, similar to [1]. In specific, given the node-level feature $\mathbf{X} \in \mathbb{R}^{|\mathcal{V}| \times m}$ where the $n$-th row, $x_n \in \mathbb{R}^m$ representing the embedding of node $n$, we firstly obtain the global context $c \in \mathbb{R}^m$ as $c = tanh(\frac{1}{N}W\sum_{n=1}^{N}x_n)$, where $W \in \mathbb{R}^{m \times m}$ is a learnable matrix. Then, there is a node-wise attention to be aware of the similarity between node and global context: $h = \sum_{n=1}^{N}\sigma(x_n^T c)x_n$ where $\sigma(\cdot)$ is the sigmoid function and $h \in \mathbb{R}^m$ is the graph-level embedding.

The concatenated feature of graph $i$ and $j$ is denoted as $h_{ij} = \text{CONCAT}(h_i, h_j) \in \mathbb{R}^{2m}$. Since features $h_i, h_j$ come from different graphs, it is necessary to weigh the importance of each for the selection of useful ones. The attention mechanism can help to explore the element-wise dependence among the features of two graphs for concatenating them smoothly in the feature space. Therefore, we apply an attention layer on the concatenated feature $h_{ij}$ to accomplish this goal as:

$$h_{ij}^* = \text{MLP}(\varphi(W_U \delta(W_D h_{ij})) \cdot h_{ij} + h_{ij}), \tag{4}$$

where $h_{ij}^* \in \mathbb{R}^d$ is regarded as the joint embedding of graph $i$ and graph $j$, and $\varphi(\cdot)$ and $\delta(\cdot)$ denote the sigmoid gating and ReLU function respectively. $W_D$ is the weight set of a NN layer, which acts as downscaling with reduction ratio $r$ assigned as 4. After ReLU activation, the low-dimension signal is then increased to $h_{ij}$ with the ratio $r$ by a upscaling layer, whose weight set is denoted as $W_U$.

As shown in Fig. 3, there is an additional EFN between the feature encoder and regression head. Such EFN is applied to fusing the multi-level joint embeddings across pairing graphs. Following the similar strategy, we firstly achieve the concatenated multi-level features $h_{ij}^{all} = \text{CONCAT}(h_{ij}^{(1)}, h_{ij}^{(2)}, h_{ij}^{(3)}) \in \mathbb{R}^{3d}$. Then, an EFN is applied to take the concatenated embedding $h_{ij}^{all}$ for multi-level feature fusion as Eq. (4): $h_{ij}^* = \text{EFN}(h_{ij}^{all}) \in \mathbb{R}^D$, where $D$ is assigned as 16.

The whole early-fusion network consists of two components: the encoder net and the regression net parameterized by $\Theta_E$ and $\Theta_R$. As shown in Fig. 3, the GIN and EFNs stated above can be summarized as an encoder net as $h_{ij}^* = E(\mathcal{G}_i, \mathcal{G}_j, \Theta_E)$. Then, an MLP-based regression net is attached to project the joint embedding $h_{ij}^*$ into the desired similarity score $s_{ij}$ optimized by the MSE loss as:

$$\mathcal{L}_{reg} = \frac{1}{|\mathcal{D}|}\sum_{i,j \in \mathcal{D}}\left(R(E(\mathcal{G}_i, \mathcal{G}_j, \Theta_E), \Theta_R) - s_{ij}\right)^2, \tag{5}$$

where $R(\cdot)$ denotes the regression network and $\mathcal{D}$ represents the set of all the training graphs.

### 3.3 Efficient Graph Similarity Computation

Although the proposed early-fusion network can achieve the competitive results with a similar time cost as previous co-attention-based methods [1, 2, 22], there are two crucial limitations on the efficiency of such methods: 1) the individual graph embeddings are unable to collect; 2) there is still a room to improve inference speed. In the paper, we have further taken the Knowledge Distillation (KD) and linear regularization for embedding decomposition to address such two challenges.

### 3.3.1 Embedding Decomposition

To decompose the joint embedding $h_{ij}^*$ into the separate individual embeddings $h_i^*$ and $h_j^*$ is a necessary step for KD. The primary reason for embedding decomposition is that we hope to achieve

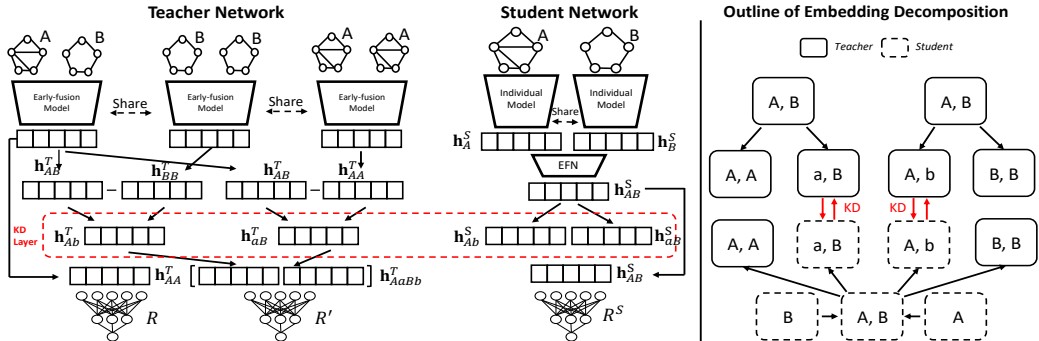

Figure 4: Illustration of embedding decomposition and KD process between the teacher and student models. The pseudo individual embeddings, which are applied for KD, are collected as the linear subtraction between joint embedding and duplicate graph embedding. More details are in Sec. 3.3.

the individual embeddings for offline storage. The other reason involves the stability of the knowledge transfer. We found that distilling the joint embeddings between the teacher and student models failed to work. More details about this point will be provided in the ablation study of Sec. 4.3. Such a phenomenon indicates the necessity to separate the individual ones from the joint embedding. Then, the individual features will be aligned between the teacher and student models through the KD loss.

The detail of the proposed linear embedding decomposition is shown in Fig. 4. The basic assumption of this design is that the joint embedding might be represented as the linear combination of individual embeddings in the high-dimensional feature space. Specifically, given graph $A$ and graph $B$, the joint embedding can be easily achieved as $h^*_{AB} = E(\mathcal{G}_A, \mathcal{G}_B)$. Moreover, we also have access to the $h^*_{AA} = E(\mathcal{G}_A, \mathcal{G}_A)$ and $h^*_{BB} = E(\mathcal{G}_B, \mathcal{G}_B)$ given duplicate inputs. Under the assumption of linear combination, the pseudo individual graph embedding will be computed as $h^*_{aB} = h^*_{AB} - h^*_{AA}$ where $h^*_{aB}$ is supposed to cover all the knowledge of graph $B$ and parts of graph $A$. And the pseudo individual graph embedding of graph $A$ is collected in the same way: $h^*_{Ab} = h^*_{AB} - h^*_{BB}$. To ensure the consistence with the desired task, we later concatenate the pairs of pseudo individual graph embeddings as $h^*_{AaBb} = \text{CONCAT}(h^*_{aB}, h^*_{Ab})$ that redundantly covers the knowledge of joint embedding $h^*_{AB}$. Another MLP-based regression network $R'$ is applied to project it into the desired target score $R'(h^*_{AaBb}, \Theta'_R) \in \mathbb{R}$ optimized by the MSE loss as Eq. ( 4):

$$\mathcal{L}'_{reg} = \frac{1}{|\mathcal{D}|} \sum_{i,j \in \mathcal{D}} \left( R'([h_{i,j} - h_{i,i}; h_{i,j} - h_{j,j}], \Theta_{R'}) - s_{ij} \right)^2, \tag{6}$$

where $h_{i,j} = E(\mathcal{G}_i, \mathcal{G}_j, \Theta_E)$, $h_{i,i} = E(\mathcal{G}_i, \mathcal{G}_i, \Theta_E)$, $h_{j,j} = E(\mathcal{G}_j, \mathcal{G}_j, \Theta_E)$ and $[\,\cdot\,;\,\cdot\,]$ represents the operator of two features concatenation. More details and the verification of the proposed linear embedding decomposition are provided in the ablation study of Sec. 4.3.

### 3.3.2 Knowledge Distillation (KD)

To get a fast model from a slow one, there are multiple compression solutions such as pruning, quantitation, etc. This paper adopts a more practical and effective method to handle this issue by using the knowledge distillation [26, 16]. As shown in Fig. 4, with the linear embedding decomposition of the joint feature $h^T_{AB}$, we could obtain pseudo individual embeddings $h^T_{aB}$ and $h^T_{Ab}$ of the teacher model. For the student model, we take a siamese GIN as the feature encoder, i.e., $h^S_A = \text{GIN}(\mathcal{G}_A, \Theta^S_E)$ and $h^S_B = \text{GIN}(\mathcal{G}_B, \Theta^S_E)$. Then, the next step is to fuse the individual embeddings to achieve the joint embedding as $h^S_{AB} = I(h^S_A, h^S_B, \Theta^S_I)$, where $I(\cdot)$ is a standard EFN. The pseudo individual embeddings, i.e., $h^S_{Ab}$ and $h^S_{aB}$, is computed following the same strategy of the teacher network.

To enforce the student model to inherit the teacher model's knowledge, it is necessary to minimize the discrepancy of the pseudo individual features. Here we apply both the first order and second order distance [26] for distillation. Therefore, the knowledge distillation (KD) loss is formulated as:

$$\mathcal{L}_{KD}(\mathcal{G}_A, \mathcal{G}_B) = \frac{\alpha}{2}(\left\| h^T_{Ab} - h^S_{Ab} \right\|_1 + \left\| h^T_{aA} - h^S_{aB} \right\|_1) + (1-\alpha)l_\delta(\psi_D(h^T_{Ab}, h^T_{aB}), \psi_D(h^S_{Ab}, h^S_{aB})), \tag{7}$$

where $\psi_D(h_i, h_j) = \left\| h_i - h_j \right\|_1$ is distance-wise potential function measuring the first order distance in the same domain, and $l_\delta$ is the Huber loss [26]. The second order distance is used to maintain the

Table 1: Quantitative GED results of baselines and our method over AIDS, LINUX, IMDB and ALKANE.

| Methods | AIDS | | | | | LINUX | | | | |
|---|---|---|---|---|---|---|---|---|---|---|
| | MSE ↓ | $\rho$ ↑ | $\tau$ ↑ | p@10 ↑ | p@20 ↑ | MSE ↓ | $\rho$ ↑ | $\tau$ ↑ | p@10 ↑ | p@20 ↑ |
| Beam | 12.09 | 0.609 | 0.463 | 0.481 | 0.493 | 9.268 | 0.827 | 0.714 | 0.973 | 0.924 |
| Hungarian | 25.30 | 0.510 | 0.378 | 0.360 | 0.392 | 29.81 | 0.638 | 0.517 | 0.913 | 0.836 |
| VJ | 29.16 | 0.517 | 0.383 | 0.310 | 0.345 | 63.86 | 0.581 | 0.450 | 0.287 | 0.251 |
| GENN-A* | **0.635** | **0.959** | - | **0.871** | - | 0.324 | **0.991** | - | 0.962 | - |
| SimGNN | 1.189 | 0.843 | 0.690 | 0.421 | 0.514 | 1.509 | 0.939 | 0.830 | 0.942 | 0.933 |
| E-SimGNN | 2.096 | 0.869 | 0.699 | 0.534 | 0.641 | 0.469 | 0.982 | 0.892 | 0.971 | 0.968 |
| GMN | 1.886 | 0.751 | - | 0.401 | - | 1.027 | 0.933 | - | 0.833 | - |
| GraphSim | 0.787 | 0.874 | - | 0.534 | - | **0.058** | 0.981 | - | 0.992 | - |
| EGSC-T | 1.601 | 0.901 | **0.739** | 0.658 | 0.729 | 0.163 | 0.988 | **0.908** | **0.994** | **0.998** |
| EGSC-S | 1.546 | 0.898 | 0.736 | 0.649 | 0.724 | 0.293 | 0.984 | 0.898 | 0.978 | 0.983 |

| Methods | IMDB | | | | | ALKANE | | | | |
|---|---|---|---|---|---|---|---|---|---|---|
| | MSE ↓ | $\rho$ ↑ | $\tau$ ↑ | p@10 ↑ | p@20 ↑ | MSE ↓ | $\rho$ ↑ | $\tau$ ↑ | p@10 ↑ | p@20 ↑ |
| SimGNN | 1.264 | 0.878 | 0.770 | 0.759 | 0.777 | 2.446 | 0.859 | 0.686 | 0.87 | 0.782 |
| E-SimGNN | 1.148 | 0.864 | 0.75 | 0.806 | 0.807 | 1.622 | 0.886 | 0.722 | 0.982 | 0.955 |
| GMN | 4.422 | 0.725 | - | 0.604 | - | - | - | - | - | - |
| GraphSim | 0.743 | 0.926 | - | 0.828 | - | - | - | - | - | - |
| EGSC-T | **0.553** | **0.938** | **0.829** | **0.872** | **0.878** | **0.533** | **0.930** | **0.787** | **0.998** | **0.991** |
| EGSC-S | 0.581 | 0.935 | 0.826 | 0.857 | 0.869 | 1.198 | 0.899 | 0.741 | 0.993 | 0.978 |

relational information. $\alpha$ is a trade-off parameter assigned as 0.5. On the top of the KD layer, an MLP-based regression network will be attached over the joint embedding $h_{AB}^S$. Apart from the KD loss, there is a supervision (i.e., MSE) loss $\mathcal{L}_{reg}^S$ on the student model to fulfill the object of the task.

## 4 Experiments

Although our proposed approach can be generalized to different graph distances, we pick the Graph Edit Distance (GED) as the evaluation task, which follows the standard protocol [1].

### 4.1 Setup

We deploy the GIN [43] as the backbone of the encoder network. The regression network is a two-layer MLP with randomly initialed weights. Our proposed method includes two versions: 1) the teacher network and the 2) student network which are denoted as **EGSC-T** and **EGSC-S**, respectively. To optimize the proposed model, we take the Adam [20] as the optimizer based on PyTorch Geometric (PyG) [31, 10]. The learning rate is assigned as 0.001 with weight decay 0.0005. The batch size is 128, and the model will be trained over 6,000 epochs. Our implementation depends on PyG-based re-implementations of SimGNN [‡] and Extended-SimGNN [§]. All experiments are run on the machine with Intel i7-5930K CPU@3.50GHz with 64GB memory.

#### 4.1.1 Benchmarks

Our proposed method has been evaluated over four popular datasets: **AIDS**, **LINUX**, **IMDB** and **ALKANE**. We have used the standard dataloader, i.e., 'GEDDataset', directly provided in the PyG [¶].

• **AIDS** (i.e., AIDS700nef) is composed of 700 chemical compound graphs which is split into 560/140 for training and test. Each graph has 10 or less nodes assigned with 29 types of labels.

• **LINUX** dataset consists of program dependence graphs generated from the Linux kernel. Each graph represents a function, where a node represents a statement and an edge means the dependency. There are 1000 graphs in total with equal or less than 10 nodes each. The nodes have no labels.

• **IMDB** dataset (i.e., "IMDB-MULTI") has 1,500 unlabeled graphs representing ego-networks of movie actors/actresses. There will be an edge if the two actors/actresses show in the same movie.

---

[‡] https://github.com/benedekrozemberczki/SimGNN

[§] https://github.com/gospodima/Extended-SimGNN

[¶] https://pytorch-geometric.readthedocs.io/en/latest/_modules/torch_geometric/datasets/ged_dataset.html#GEDDataset

Table 2: Ablation study results over the AIDS and IMDB datasets. **KD** represents the knowledge distillation.

| Methods | KD | AIDS | | | | | IMDB | | | | |
|---|---|---|---|---|---|---|---|---|---|---|---|
| | | MSE $\downarrow$ | $\rho \uparrow$ | $\tau \uparrow$ | p@10 $\uparrow$ | p@20 $\uparrow$ | MSE $\downarrow$ | $\rho \uparrow$ | $\tau \uparrow$ | p@10 $\uparrow$ | p@20 $\uparrow$ |
| w/o Attn | ✗ | 1.762 | 0.899 | 0.737 | 0.651 | 0.724 | 0.752 | 0.933 | 0.823 | 0.856 | 0.868 |
| w/o GIN | ✗ | 2.158 | 0.863 | 0.691 | 0.535 | 0.637 | 0.594 | 0.926 | 0.803 | 0.862 | 0.866 |
| Single Level | ✗ | 1.824 | 0.875 | 0.706 | 0.576 | 0.658 | 0.690 | 0.930 | 0.815 | 0.850 | 0.865 |
| Student | ✗ | 1.770 | 0.882 | 0.717 | 0.601 | 0.683 | 0.763 | 0.928 | 0.813 | 0.829 | 0.851 |
| Teacher | ✗ | **1.601** | **0.901** | **0.739** | **0.658** | **0.729** | **0.553** | **0.938** | **0.829** | **0.872** | **0.878** |
| Joint Feat | ✓ | 2.258 | 0.874 | 0.703 | 0.588 | 0.679 | 1.032 | 0.872 | 0.761 | 0.814 | 0.829 |
| 1st Order | ✓ | 1.604 | 0.894 | 0.731 | 0.614 | 0.715 | **0.548** | 0.934 | 0.824 | 0.856 | 0.865 |
| 2nd Order | ✓ | 1.647 | 0.893 | 0.731 | 0.631 | 0.715 | 0.692 | 0.929 | 0.814 | 0.847 | 0.866 |
| w/o $\mathcal{L}'_{reg}$ | ✓ | 1.711 | 0.890 | 0.726 | 0.612 | 0.710 | 0.694 | 0.926 | 0.811 | 0.842 | 0.860 |
| Student | ✓ | **1.546** | **0.898** | **0.736** | **0.649** | **0.724** | 0.581 | **0.935** | **0.826** | **0.857** | **0.869** |

Table 3: Inference time to solve GED computation on AIDS. Student-R means the student model with raw input graphs. Student-F denotes that the embeddings are stored offline, which can be online loaded for inference.

| Model | GENN-A* | SimGNN | E-SimGNN | E-SimGNN-F | Teacher | Student-R | Student-F |
|---|---|---|---|---|---|---|---|
| Time | 290.1h | 11.139s | 9.672s | 3.464s | 11.139s | 10.149s | **0.148s** |

• **ALKANE** [4] is a purely structural dataset containing 120 chemical compound graphs. All the graphs are acyclic (i.e., trees) without node labels. There is no split of training and testing in the PyG.

### 4.1.2 Evaluation Metric

**Mean Squared Error (MSE)** (in the format of $10^{-3}$) is the most popular matrix that measures the average squared error between the predicted scores with the ground-truth similarities. **Spearman's Rank Correlation Coefficient ($\rho$)** and **Kendall's Rank Correlation Coefficient ($\tau$)** evaluate the correlation of ranking-wise computed results and ground-truth results. **Precision at k (p@k)** is the intersection of top k predicted results with the ground-truth top k over the value k.

### 4.1.3 Baselines

**Beam** [27] is a variant of the A* algorithm [14] in sub-exponential time by beam search. **Hungarian** [33] is the cubic-time algorithm based on the Hungarian Algorithm for bipartite graph matching, and the **VJ** [9] algorithm is a variant of Hungarian method. **SimGNN** [1] is a co-attention-based GSC method that directly predicts the GED score given two input graphs. **Extended-SimGNN**[§] (i.e., E-SimGNN) is an improved version of SimGNN using GIN as the backbone. **GraphSim** [2] is a multi-scale model which fuses the cross-graph features in multiple GNN layers. **GMN** [22] is another GNN-based method. It manages to fuse the cross-graph information with the node-level message passing. **GENN-A*** [40] is the more recent work which applies the GNN to accelerate the hard GED solvers such as A*. Beam, Hungarian, VJ and GENN-A* are the GED solvers that require to output edit path, which, however, are hard to generalize to other GSC metrics. Most of the baseline results are copied from their published papers, and we run the Extended-SimGNN for results collection.

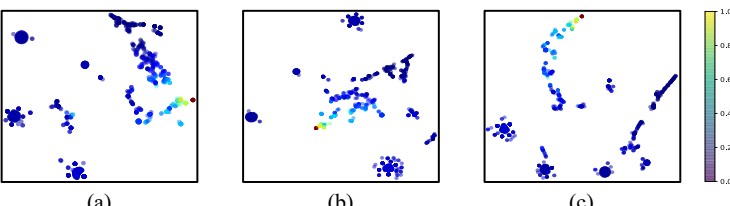

| (a) | (b) | (c) |

Figure 5: t-SNE Visualization of joint embeddings on IMDB. (a)-(c) SimGNN; Extended-SimGNN; Our Teacher Model. The color of dots represent the similarity score decreasing from 1 to 0.

### 4.2 Quantitative Results

The quantitative results on GED are summarized in Tab. 1. The results of SimGNN on ALKANE are run by us. It is easily observed that the proposed methods, including both the early-fusion model (i.e., teacher model) and student model, outperform the baselines on most of the scenarios. Although ours

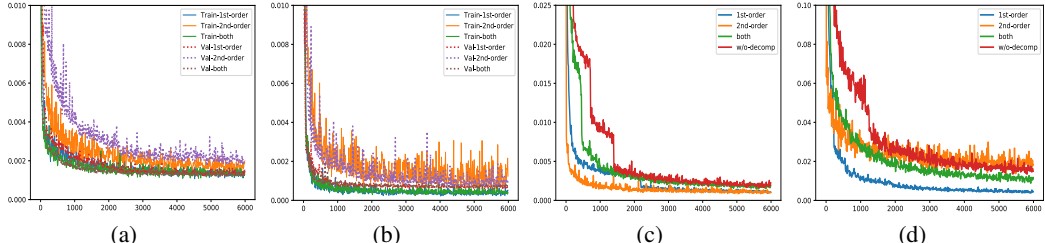

Figure 6: The curve of losses through KD. (a)-(d): Training and validation MSE loss on AIDS; Training and validation MSE loss on IMDB; KD loss on AIDS; KD loss on IMDB;

are beaten by GENN-A* in some cases, the proposed approaches have the superiority in extensibility and scalability since there is no need to output the edit path step-by-step. On the IMDB and ALKANE datasets, the teacher model obviously outperforms the baseline ones with a large margin. Comparing the performance of teacher and student model, there is a slight superiority of the former one in most of the cases. While the student model beats the teacher on the MSE metric of the AIDS dataset, which means the reducing model redundancy can further improve the performance in some cases.

### 4.3 Ablation Study

To investigate the effects of each module, we introduce the ablation study on the AIDS and IMDB datasets in Tab. 2 and provide some visualization results in the Subsection 4.5. As shown in Tab. 2, the w/o KD setting has five different components including: without attention in EFN; taking GCN as the backbone (i.e., w/o GIN) to analyze the effects of GIN; Single Level meaning only taking the final-layer GIN feature for embedding fusion; Student and Teacher. It is reasonable to compare the student models with or without KD. By comparing such two results, the with-KD model has a strong superiority over the latter one. And we can easily find that the teacher model should be regarded as the upper bound of the with-KD student model. Considering the process of KD, the embedding decomposition proves to be useful since the joint feature KD (i.e., Joint Feat) is largely inferior to the pseudo individual one. Although there is no much difference between the first-order and second-order distances, the combination of such two distances is helpful to boost the overall performance.

### 4.4 Inference Time

The comparison on inference time on the whole testing set is shown in the Tab. 3 where the Hungarian and GENN-A* are copied from [40] and others are run by our own. In the Tab. 2 of [40], whose reported time is for one pair of graphs, we have transformed such a time into the whole testing set time as: $OnePairTime \times DataNumber$, i.e., $290.1h = 29.915s \times 78400$. The student model beats other methods in **two orders** in the case of embedding-based inference. Such results sufficiently indicate the high efficiency of our siamese-based student model in the GSC task, which has the potential for real-time setting.

### 4.5 Analysis and Visualization

**Convergence Analysis.** We evaluate the convergence of baseline methods as well as our proposed methods on the ablation scenarios in Fig. 6. Comparing the sub-figures (a) and (b), we can clearly see that the proposed method (i.e., 'both') reaches the lower MSE loss through iteration. And the models turn to converge after $5,000 - 5,500$ epochs which makes $6,000$ a good choice. Moreover, the Val-both loss is highly overlapped with the training loss (i.e., 'Train-both'), which means that there is no clear overfitting of our models. In the KD case, the second-order loss is harder to minimize.

**Feature t-SNE Visualization.** As illustrated in Fig. 5, we employ the t-SNE algorithm [37] to visualize joint embeddings obtained by the encoder given a fixed query graph. The features learned by our approach are more clustered and separable in comparison with (a) and (b).

**Example Ranking Results.** As shown in Fig. 7, there are no clear differences and errors in the top 5 ranking results. While, the baselines fail to rank the correct graphs in the later sequence, which indicates the superiority of our teacher model in handling the more challenging cases.

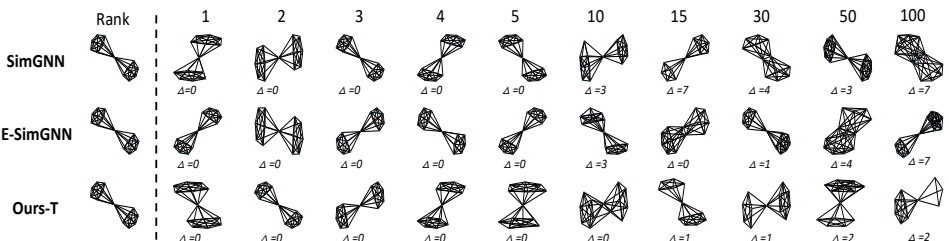

Figure 7: Ranking results of SimGNN, E-SimGNN and our teacher model on IMDB. $\Delta$, i.e., $\Delta = |GED_{pred} - GED_{gt}|$, represents the absolute difference between the ground truth GED and the GED of predicted result.

## 5 Conclusion

This paper proposes a novel GSC approach for **fast inference** based on the **slow learning**. The slow learning involves designing a co-attention-based feature fusion network on multilevel GNN features that achieves cutting-edge accuracy. To further accelerate the inference speed without much accuracy drop, we apply the knowledge distillation to compress the proposed co-attention network, i.e., teacher model, to the student one. Moreover, such a student model also enables the offline collection of individual graph embeddings, which is beneficial for online retrieval. We decompose the joint embedding into the pseudo individual ones linearly for precise teacher-student alignment to address the instability through knowledge transfer. The experiments on four real-world datasets demonstrate our approach's superiority over the previous methods on both accuracy and efficiency.

**Acknowledgments**. Thanks to the grant from Adobe Research to support this project.

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
