# OpenReview forum: "Slow Learning and Fast Inference: Efficient Graph Similarity Computation via Knowledge Distillation"
_NeurIPS.cc/2021/Conference — NeurIPS 2021 Poster_

### Official Review · Reviewer_9hgj · 2021-07-02

**Rating:** 6
**Confidence:** 3

**Summary:**

This paper tackles the problem of graph similarity computation (GSC). The key insight of this paper is that to better model the problem of GSC, early and extensive graph feature fusion is needed, which is harmful to model efficiency. To address this issue, the authors propose a knowledge distillation solution that drops all early fusions. This allows the regression models to take in only the embeddings, which can be pre-computed, and thus saves time. Experimental evaluations indicate that the proposed framework achieves good performance and is fast in inference.

**Ethical Concerns:**

No.

**Limitations And Societal Impact:**

They are properly addressed.

**Main Review:**

Overall I like this paper. The paper is based on insightful observations about feature fusion and efficiency, and the proposed technical solution, although not completely new, is sound. The experimental evaluations are extensive. `

**Strengths**:
- **Clarity**: The paper is in general easy to follow.
- **Experiments are extensive**. I specifically appreciate the extensive ablation studies on the design.
- **Sound technical solution**. The technical solutions are based on insightful and realistic observations.

**Weaknesses**:
- **Somewhat lack in novelty**: As attention, multi-level graph feature fusion, and KD are all existing techniques, the technical solution seems a little short of novelty, although indeed, it is technically sound.
- **Unfair comparison of inference time**. Since the query graphs may come at any time and may be arbitrary new graphs, I think the comparison of inference time in Table 3 is not very fair, as for completely new query graphs, you still need to first compute its embedding.

**Minor comments**:
- Line 173, 'a node-wise attention to aware of...', to aware of -> to be aware of
- Line 174, $h = \sum_{n=1}^N \sigma(x^T_ncx_n)$, should it be $\sigma(x^T_nc) x_n$?
- The notations through Sec. 3.2.2 are confusing. For example, $h_{ij}^{all}$ and $\mathbf{h}_{ij}^{all}$ are interleavingly used.  Please consider unifying them.
- Line 220-221, consider revising the notations, as the $E(\cdot, cdot, cdot)$ repeats too many times.
- Eqn 7, I suppose that Huber loss is a binary function, but why is $l_\delta(\phi_D(h_{Ab}^T, h_{aB}^T), h_{Ab}^S, h_{aB}^S)$?
- Sec. 4.1.2, "matrix"->"metric".


**Time Spent Reviewing:**

4

---

> ### Author Response · Authors · 2021-08-10
> **Author Response to Reviewer 4 (9hgj)**
>
> Thank you very much for the valuable comments. Your concerns and questions are addressed below:
>
> **Q1**: Somewhat lacking in novelty.
>
> **AQ1**: Thanks for your comments. We totally agree that part of our model is similar to existing approaches, and we would appreciate you for letting us explain more about the novelties. Fast computation of graph similarities is the major goal of this paper. To this end, one of the main technique novelties of this paper is to propose a solution on the KD over the joint features. There are few papers on this topic in the Vision or NLP domains. However, to the best of our knowledge, this is the first work on the graph domain, and previous works fail to work on such cases. We propose a simple but novel way to decompose the joint feature for knowledge transfer. Moreover, offline feature collection is another contribution of this paper. Due to the successful KD, it enables a tiny student model directly inputting embeddings for GSC which has been greatly accelerated. We would appreciate you for considering these as our model novelties.
>
>
>
> **Q2**: Unfair comparison of inference time considering that the new query graph needs computing embedding.
>
> **AQ2**: This is a very good question. In Table 3, we have already assumed that the query graphs are not seen before, and their features are not offline stored. The new query graphs are coming in batch-wise. So, the inference time of Student-F includes computing the features of new query graphs. We will explain this in the following version paper.
>
>
> **Minor Comments**:
>
> **Answer**: We sincerely appreciate the reviewer for providing such comments on formulation, writing, and grammar. The  $\sigma(x_n^{T}cx_n)$ in line 174 should be revised as $\sigma(x_n^{T}c)x_n$. In Eqn. 7, the Huber loss is applied to measure the distance of distance, which we referred to as the 2nd order distance. All the other issues are corrected in the latest draft, and we will continue polishing the paper until publishing.

---

> > ### Comment · Reviewer_9hgj · 2021-08-18
> > **Thanks for the reply.**
> >
> > Thanks for the reply. I now better understand the contributions of this paper and the comparison setting.

---

### Official Review · Reviewer_Srnx · 2021-07-16

**Rating:** 6
**Confidence:** 4

**Summary:**

The paper presents a new method for computing graph similarity efficiently and approximately using graph neural networks. A novel attention-based neural network is proposed for combining the embeddings for the two input graphs. A novel way to compute graph similarity faster is proposed based on embedding decomposition and applying knowledge distillation to obtain a fast model from a slow model. Experiments on four graph datasets demonstrate the effectiveness and efficiency of the proposed method.

**Limitations And Societal Impact:**

There are several things that could further improve the quality of the work:

1. The authors use four benchmark datasets, but given the high standard of NeurIPS and the relatively small sizes of the four datasets, I strongly recommend the authors using larger datasets in terms of both the number of graphs and the average number of nodes. Specifically, the authors do not mention the detailed statistics of the datasets, but by checking with previous works, it seems the largest dataset contains a few hundreds of nodes on average. Scaling to graphs of 1K, 10K, 100K, and even 1M nodes will show the strengths and potential limitations of the proposed method better. This is especially important since one key contribution is the fast model learned through embedding decomposition and knowledge distillation.

The issue of dataset being small is also seen from Section 4.4 and Table 3, where the authors show the average inference time on AIDS. AIDS only contains graphs of <= 10 nodes, and using this dataset to demonstrate how fast the student model runs is fine, but if using a very large graph dataset of more nodes on average, the speed-up gained by the fast model can be more significant.

2. The authors mention several graph metrics in Section 2.1, but only GED is used in the experiments. The authors should consider testing the methods on other metrics such as MCS.


**Main Review:**

Originality: The task addressed in the paper is graph similarity computation (GSC), which can incorporate several hard-to-compute metrics such as GED, MCS, etc. The proposed method contains a novel Embedding Fusion Network (EFN) and a novel way to apply knowledge distillation to the task of GSC. It is clear how the proposed model is different from previous works, and the important previous works are adequately cited and compared against in the experiment.

Quality: The submission is technically sound with claims relatively well supported by experimental results. The authors also list several limitations of the work in the supplementary material. However, I recommend the authors testing the methods on larger graph datasets and more graph similarity metrics to better support the claims.

Clarity: The submission is clearly written and well organized.

Significance: The paper indeed points out a potentially very promising direction of computing graph similarity in an efficient way, ie using knowledge distillation. The idea of learning both a slow model and fast model on its own is not new but is a very interesting and promising direction, and is relatively underexplored in the graph deep learning domain. I can foresee future researchers could build upon this idea.




**Time Spent Reviewing:**

4

---

> ### Author Response · Authors · 2021-08-10
> **Author Response to Reviewer 3 (Srnx)**
>
> Thank you very much for the valuable comments. Your concerns and questions are addressed below:
>
> **Q1**: More statistics of the datasets?
>
> **AQ1**: Thanks for your question. We have calculated 1) the number of graph pairs in the test set, 2) the average quantity of nodes per graph, and 3) the average number of edges per graph of four used benchmarks. Here is the detailed information listed below.
>
> |  | Quantity of Graph Pairs | Avg Nodes per Graph | Avg Edges per Graph |
> |:---:|:---:|:---:|:---:|
> | AIDS700nef | 78,400 | 8.91 | 17.675 |
> | LINUX | 160,000 | 7.59 | 13.92 |
> | IMDBMulti | **360,000** | **13.66** | **143.49** |
> | ALKANE | 14,400 | 8.88 | 15.77 |
>
> Table 1. Statistics of the testing sets of four benchmarks.
>
> **Q2**: Will the proposed method be evaluated on the larger GSC benchmarks?
>
> **AQ2**: That is an excellent question. As mentioned in the introduction of the paper, it is quite challenging to calculate the graph similarities in terms of the computation cost, especially for the exact solvers, which is an NP-hard problem. There is a dilemma that the ground truth GED between the large graphs is often inaccessible due to the intolerant computation cost. Manual labeling is also extremely hard since it is GED between large graphs cannot be directly observed by human efforts. We have surveyed a lot on the large GED datasets and found most of them are around 10 nodes per graph. This is a promising direction to which our proposed method will be extended. As shown in the Table 1 in response to Reviewer zANd, our proposed model is 42x faster than SOTA on the IMDBMulti, which is the largest and most challenging dataset of the four. Such results can partially support the superiority of our model on large datasets.
>
>
> **Q3**: Why only reporting the inference time on the AIDS dataset?
>
> **AQ3**: Thanks for your suggestion. To demonstrate the superiority of our proposed method on the inference speed, we further calculate the computation time on all the four datasets, which are summarized in the Table 1 of the response to the Reviewer zANd.
>
> **Q4**: The authors should consider testing the methods on other metrics such as MCS.
>
> **AQ4**: Thanks for this suggestion. The MCS is another important matrix on the GSC problems. I am already working on it and plan to add it to the next version PyG for convenient use of the future GSC methods.

---

> > ### Comment · Reviewer_Srnx · 2021-09-15
> > **Thanks for your reply.**
> >
> > Thank you for your response. I have taken your reply into considerations.

---

### Official Review · Reviewer_zANd · 2021-07-16

**Rating:** 6
**Confidence:** 3

**Summary:**

In this paper authors proposed a method to solve subgraph similarity problem using graph neural network. They proposed a knowledge distillation based technique to distill knowledge from a slow early fusion based model to a faster student model which improves the speed of the inference.


**Ethical Concerns:**

no ethical concerns

**Limitations And Societal Impact:**

Did not find the section

**Main Review:**

Originality:

The work is a novel combination of techniques like Graph Isomorphism Network, slow Early Fusion model, faster Inference model, Knowledge Distillation between the slow model and the fast model. I specially liked the design of the two models for knowledge distillation.

Quality:

The submission seems technically sound. Claims are generally well supported. One analysis I missed was, why the GENN-A* outperforms this model by a lot specially in AIDS dataset. Any intuitive analysis about this outperformance would have made the paper stronger. I still feel this direction of work is important, but it would be good to know whats missing in the work to make the work better.

I liked the ablation analysis of different parts of the model,

Also Inference time improvement is impressive, but can we improve the accuracy more even if we use similar times to the other methods especially GENN-A*. I feel like we need to analyze the trade off between time and accuracy in this work. Especially can we design a more accurate teacher model which can use more time/memory to be competitive with GENN-A*.

Clarity:

The submission is clearly written and seems well organized. There are some typo like "graph convectional network" in line 41. But the work was easy to read and understand.

Significance:

I think this is a very important problem to solve in this area. It should be impactful work for researchers in this domain. Even though it does not beat the state-of-the-art in all cases, its still a novel way to model the problem.



**Time Spent Reviewing:**

2

---

> ### Author Response · Authors · 2021-08-10
> **Author Response to Reviewer 2 (zANd)**
>
> Thank you very much for recognizing our work and the great comments. Please see our response to the concerns and questions:
>
> **Q1**: Why does the GENN-A* outperform this model by a lot, especially in the AIDS dataset?
>
> **AQ1**: Thanks for this question. The GENN-A* is a Combinatorial Optimization (CO) based method that requires the node-wise search over the tree-structure space. The GENN-A* applies a GNN to prune the irrelevant nodes on the tree whose searching space is greatly reduced to accelerate the searching process. AIDS dataset has less quantity of graphs compared with the LINUX. In GENN-A*, the GNN is applied to predict the node-wise score, requiring more fine-grained learning over the graphs. The GNN is trained by the loss average over a batch of samples, which is not totally reliable since it might be over-smoothing due to the large quantity of non-fine-grained data.
>
>
> **Q2**: Can we improve the accuracy more even if we use similar times to the other methods, especially GENN-A*?
>
> **AQ2**: We could use other techniques to improve accuracy. As mentioned above, the GENN-A* is a CO-based method whose focus is to predict graph editing paths with the node-wise computation. Therefore, the CO-based methods cannot be accelerated by the KD methods because the major time cost is in the searching. Our proposed method follows the style of SimGNN that directly takes graph-wise representation using GNN for the approximate estimation of GED score. This is a totally different direction that seems cannot be combined with the CO-based techniques. While we do have some other ideas to improve the accuracy of our proposed method, such as exploring the deeper architecture of the teacher model and designing the stronger KD loss for better knowledge transfer.
>
>
> **Q3**: Explain the trade-off between time and accuracy.
>
> **AQ3**: Thanks for this question. The trade-off between accuracy and speed is always one of the core topics in the graph and ML community. First of all, we apologize that we have misunderstood Table 2 of the original GENN-A* paper [1]. After I carefully checked its official code [2], which was just released on Aug 4th, and contacted the author for double-checking afterward, __the reported computation time of Table 2 in GENN-A* paper is on the one pair of graphs, whereas our reported time in the original Table 3 is on the whole testing set__. Therefore, for a fair comparison, the Table 3 of our paper is updated as below.
>
> |  | Hungarian-A* | GENN-A* | SimGNN | SimGNN-F | E-SimGNN | E-SimGNN-F | Teacher | Student-R | Student-F |
> |:---:|:---:|:---:|:---:|:---:|:---:|:---:|:---:|:---:|:---:|
> | AIDS700nef | 29.915 | 13.323 | 1.420e-04 | 3.305e-05 | 1.234e-04 | 4.418e-05 | 1.421e-04 | 1.295e-04 | **1.888e-06** |
> | LINUX | 2.332 | 2.177 | 2.477e-04 | 2.458e-05 | 2.365e-04 | 3.811e-05 | 2.471e-04 | 2.369e-04 |  **2.144e-06** |
> | IMDBMulti | - | - | 5.359e-04 | 7.326e-05 | 6.456e-04 | 1.475e-04 | 6.976e-04 | 7.474e-04 |  **1.753e-06** |
> | ALKANE | - | - | 2.663e-04 | 5.653e-05 | 2.469e-04 | 7.160e-05 | 3.067e-04 | 2.657e-04 |  **5.347e-06** |
>
> Table 2. Average inference time (sec) to solve GED problems on **one pair of graphs** of four different benchmarks. The meanings of SimGNN-F and E-SimGNN-F can be referred in the AQ2 to Reviewer WTSt.
>
> As shown in the table above, it is obvious the GNN-based methods have shown a significant superiority over the speed compared with GENN-A* and A* methods. Although the GENN-A* achieved promising results on AIDS, its inference speed is greatly sacrificed for a marginal gain in accuracy. It is obvious that our proposed method has a better trade-off considering both accuracy and speed.
>
>
> **Typo and Grammar Error**:
>
> **Answer**: Thanks for pointing out the typo. We have just revised it, and I will continue polishing the paper writing to make it better.
>
>
> **Limitation and Social Impact**:
>
> **Answer**: Thanks for your reminding. We have mentioned the limitation and potential negative social impact in the last section of the supplementary material. We will move this section to the main paper in the following version.
>
>
>
> [1] Wang, Runzhong, et al. "Combinatorial learning of graph edit distance via dynamic embedding." CVPR-21.
>
> [2] https://github.com/Thinklab-SJTU/GENN-Astar

---

### Official Review · Reviewer_WTSt · 2021-07-16

**Rating:** 6
**Confidence:** 3

**Summary:**

The main contribution of the paper is to propose a new teacher-student framework for efficient graph similarity computation. Different from existing early-fusion models that directly output a similarity score for two input graphs, the proposed framework uses such model as a teacher to teach a student model for fast embedding training and efficient graph similarity computation.

**Limitations And Societal Impact:**

Please give more details on inference time comparison because it is a key experiment.

**Main Review:**

Pros:

- The teacher-student framework for graph similarity computation has several advantages. First, it can collect the individual graph embeddings from the joint embeddings of the two graphs. Second, the student model can distill knowledge from the pretrained teacher model and capture the well learned embedding distance in the teacher model. Third, the teacher-student framework prevents pre-computing the graph embeddings. Overall, the proposed framework is practical and technically sound. The student model can achieve a trade-off between speed and accuracy for graph similarity computation.

- The presentation is good and the paper is easy to follow. Experiments on four benchmark datasets demonstrate the effectiveness of the proposed model.


Cons:

- In my view, some contributions of this paper are over-claiming. The authors say that they "speed up the prior art by 65x on the benchmark AIDS data." According to the results in Table 3,  it is 9.672/0.148 = 65.35. However, the 0.148s is only the inference time and does not include the time for pretraining the teacher net. It may not be fair to compare the inference time of the offline-stored student model against other early-fusion models. Besides, the reported data for the baselines *Hungarian* and *GENN-A* seems wrong. As reported in Table 3 of the GENN-A paper, their inference time is less than 1s rather than the reported more than ten seconds. So, I think the experimental results cannot well support the claim that the proposed model can speed up 65x. Perhaps, I have some misunderstandings here. So, I would be grateful if the authors could give more explanations to clarify this issue.

Typos:

- Line #126: "In specific, the edge linking the a pair of nodes including..." -> "In specific, the edge linking a pair of nodes including..."

-- After rebuttal --

- Thanks for the responses to address my concerns regarding the inference time.
- I would like to leave my score unchanged.

**Time Spent Reviewing:**

3

---

> ### Author Response · Authors · 2021-08-10
> **Author Response to Reviewer 1 (WTSt)**
>
> Thanks to the reviewer for the recognition of our paper. We sincerely appreciate your valuable comments. Your concern is addressed below.
>
> **Q1**: The inference time of GENN-A* in our Table 3 is different with the one reported in the GENN-A* paper [1].
>
> **AQ1**: Thanks for your question. In the GENN-A* paper [1], there are two tables about inference time: 1) time on solving GED problems, and 2) time on predicting h(p), respectively. The former is reported in Table 2, and the latter is shown in Table 3. We refer to Table 2 of its paper for comparison since our reported time is about solving the GED problem. More details about the comparison of inference time are explained in the AQ3 to Reviewer zANd.
>
> **Q2**: It may not be fair to compare the inference time of the offline-stored student model against other early-fusion models.
>
> **AQ2**: Thanks for your comments. The early-fusion models fail to collect features offline after the early-fusion layers, and this is the task our model proposed to address. Following your suggestion, we have added the two baseline methods (i.e., SimGNN-F and E-SimGNN-F), which can directly input embeddings extracted from a pre-trained GNN (before the early-fusion layers). In the latest draft, we revised Table 3 and included the results of such baselines. All the experiment settings and environments exactly follow the original Table 3. Here is the time on the whole test set of AIDS.  For more comparisons over four datasets on the speed, please refer to the Table 1 of AQ3 to Reviewer zANd.
>
> |  | SimGNN | SimGNN-F | E-SimGNN | E-SimGNN-F | Teacher | Student-R | Student-F |
> |:---:|:---:|:---:|:---:|:---:|:---:|:---:|:---:|
> | Time (sec) | 11.136 | 2.591 | 9.672 | 3.464 | 11.139 | 10.149 | 0.148 |
>
> Tab 1. Inference time (sec) to solve GED problems on the **whole test set of AIDS**.
>
> **Q3**: And what is the time cost of teacher pre-training?
>
> **AQ3**: Thanks for this question. To train a teacher network, each epoch takes ~0.0256 secs on the AIDS. This is slightly slower than the SimGNN, which is ~0.0167 secs per epoch.
>
> **Q4**: Typos on the paper.
>
> **AQ4**: Thanks for your careful observation. We have already corrected it in the latest draft, and we will continuously polish the paper until publishing.
>
> [1] Wang, Runzhong, et al. "Combinatorial learning of graph edit distance via dynamic embedding." CVPR-21.

---

### Decision · Program_Chairs · 2021-09-28

**Decision:**

Accept (Poster)

**Comment:**

This paper proposes a new teacher-student framework for efficient graph similarity computation. The student model decouples the interaction of two graphs learned in teacher model into two separate representation vectors, which can speed up the inference significantly. The authors’ rebuttal is general well written, and has provided new experimental results requested by the reviewer.  The reviewers have concerns on the novelty, as the main techniques are adapted from existing work. But the application of knowledge distillation for similarity computation itself is novel, and the design is smart and suits the inference need very well. The results are significant. Therefore, I recommend acceptance.

**Consistency Experiment:**

NeurIPS has a long history of experimentation. In 2014, NeurIPS ran an experiment in which 10% of submissions were reviewed by two independent committees to quantify the randomness in the review process. This year, we repeated a variant of this experiment to see how the quality of the review process has changed over time.  This paper was part of the experiment and was therefore assigned to two committees (consisting of reviewers, an Area Chair, and a Senior Area Chair) that reached independent decisions.  If both committees made the same recommendation, this recommendation was followed. If a single committee recommended acceptance, the paper was accepted (with the exception of a few cases in which the other committee identified what we considered a fatal flaw, e.g., an error in a key result).

Both committees reached the same decision: **Accept (Poster)**

The other committee assigned to the paper recommended **Accept (Poster)**.  You can find the other set of reviews, along with any follow up discussion with the authors here:
https://openreview.net/forum?id=kAFq29tuVw0